# Conjugative Plasmids Disseminating CTX-M-15 among Human, Animals and the Environment in Mwanza Tanzania: A Need to Intensify One Health Approach

**DOI:** 10.3390/antibiotics10070836

**Published:** 2021-07-09

**Authors:** Caroline A. Minja, Gabriel Shirima, Stephen E. Mshana

**Affiliations:** 1School of Life Sciences, Department of Global Health and Biomedical Sciences, Nelson Mandela African Institution of Science and Technology, Arusha 23306, Tanzania; gabriel.shirima@nm-aist.ac.tz; 2Department of Microbiology and Immunology, Catholic University of Health and Allied Sciences, Mwanza 33109, Tanzania; mshana72@bugando.ac.tz

**Keywords:** conjugation, CTX-M-15, replicon, plasmid, non-beta lactam antibiotics, One Health

## Abstract

Background: Globally, *bla*_CTX-M-15_ beta-lactamases are the most popular extended spectrum beta-lactamase alleles that are widely distributed due its mobilisation by mobile genetic elements in several compartments. We aimed to determine the conjugation frequencies and replicon types associated with plasmids carrying *bla*_CTX-M-15_ gene from Extended Spectrum Beta-lactamase producing isolates in order to understand the dissemination of resistance genes in different compartments. Material and methods: A total of 51 archived isolates carrying *bla*_CTX-M-15_ beta-lactamases were used as donors in this study. Antibiotic susceptibility tests were performed as previously described for both donors and transconjugants. Conjugation experiment was performed by a modified protocol of the plate mating experiment, and plasmid replicon types were screened among donor and transconjugant isolates by multiplex Polymerase Chain Reaction in a set of three primer panels. Results: The conjugation efficiency of plasmids carrying *bla*_CTX-M-15_ was 88.2% (45/51) with conjugation frequencies in the order of 10^−1^ to 10^−9^ and a 100% transfer efficiency observed among *E. coli* of animal origin. Majority of donors (*n =* 21) and transconjugants (*n* = 14) plasmids were typed as either Inc FIA or Inc FIB. Resistance to non-beta-lactam antibiotics was transferrable in 34/45 (75.6%) of events. Ciprofloxacin, tetracycline and sulphamethoxazole-trimethoprim resistance was co-transferred in 29/34 (85.3%) such events. Gentamicin resistance was transferred in 17/34 (50%) of events. Conclusions: Majority of plasmids carrying *bla*_CTX-M-15_ were conjugatively transferred by IncF plasmids along with non-beta lactam resistance. There is a need for more research on plasmids to understand how plasmids especially multi replicon plasmids interact and the effect of such interaction on conjugation. One Health approach is to be intensified to address antimicrobial resistance which is a public health threat.

## 1. Introduction

The increasing trend of antimicrobial resistance is intensified by mobile genetic elements that harbour resistance genes [1]. The effect of these elements is extensively reported among bacteria of the *Enterobacteriaceae* family where multi drug resistance (MDR) is high. The CTX-M extended-spectrum beta-lactamases (ESBL) are the most successful MDR determinants [2], with over 100 alleles in five distinct phylogenetic groups [3]. The ecological success of CTX-M-ESBL attributes to the enzymes’ spread both clonally and horizontally [3,4,5] in multiple hosts that include *Acinetobacter* spp., *Enterobacter* spp., *E. coli*, *P. aeruginosa*, *K. pneumoniae* and *P. mirabilis*.

In natural environments, ESBL enzymes are chromosomally mediated by the selection pressure induced by beta-lactamase-producing soil organisms [6,7] or the irrational use of third-generation cephalosporins [6,8], however as previously reviewed [9], plasmid-mediated ESBL resulted from transposon-mediated insertion of different *bla*_CTX-M_ genes in bacteria chromosome. Specifically, the precursors of plasmid-mediated *bla*_CTX-M-15_ are environmental *Kluyvera* spp. whose chromosomal CTX-M clusters are incorporated into the chromosome of host bacteria by mobilising elements such as *ISEcp1* or *ISCR1.* Moreover, the location of *ISEcp1* upstream *bla*_CTX-M_ genes together with multiple inverted repeats downstream the gene facilitates the expression and ongoing transposition of *bla_KLU_* genes that result to various CTX-M enzymes, including plasmid-mediated *bla*_CTX-M-15_ [9,10]. The mobilisation potential of *ISEcp1* for chromosome-linked multi-resistant determinants in other members of *enterobacteriaceae* increases with the additional possession of *ISCR1*, another mobile genetic element (MGE) embedded in a Class 1 integron that mobilises unrelated CTX-M groups in similar or different species.

As vectors and carriers of AMR genes, plasmids are responsible for the intracellular accumulation and intercellular transfer of these genes by the process of conjugation [11]. In such cases, high conjugation rates ensure the stable long-term persistence of plasmids and associated AMR genes in minimal fitness costs even in the absence of selection pressure [12,13,14]. A multidrug resistance phenomenon is observed when these plasmids are associated with other MGE possessing different resistance determinants and code for adaptive traits such as virulence or metal resistance genes among host bacteria strains [15,16].

The globally disseminated O25: H4-ST131 *E. coli* clone producing CTX-M-15 is by conjugative IncF plasmids that are frequently recovered from hospital and community settings [17]. In Tanzania, the prevalence of bacteria producing ESBL ranges between 25 and 50 percent [18], with *bla*_CTX-M-15_ as the predominant allele in both community [19] and hospital settings [20]. The gene is also observed among companion and domestic animals and the environment combined with quinolone and aminoglycoside resistance genes [21,22]. Therefore, as demonstrated by its discovery in a novel *Enterobacter* spp. [23] and location in multiple plasmid types such as IncF, IncY and IncHI1, there is a possibility of an extensive variation in the epidemiology of *bla*_CTX-M-15_ carrying plasmids in Tanzania.

The presence of *bla*_CTX-M-15_ gene in multiple *E.coli* clones of human, animal and the environment of Tanzania [24], and limited information on the persistence of the gene’s alleles in any compartments can lead to the acquisition, transmission and evolution of new resistant strains even among non-conjugative bacteria. Since plasmids facilitate the spread of AMR genes in different compartments efforts to understand their spread and establishment in these settings is unquestionable. This study has improved our understanding of the importance of IncF plasmids in disseminating multidrug-resistant determinants among human, animal and environmental settings. It has further highlighted the importance of collaborative One Health based efforts that focus on animal and human health as critical when addressing the global threat.

## 2. Results

### 2.1. Isolates Characteristics

*Escherichia coli* was the only species isolated in both human and animals. The environment included isolates from soil and fresh water fish and comprised of *E. coli, K. pneumoniae, C. braakii and E. cloacae* species (Table 1).

### 2.2. Conjugation Efficiency of bla_CTX-M-15_ Gene among Isolates of Human, Animals and the Environment

Among 51 *bla*_CTX-M-15_ positive donor isolates, 45 (88.2%) transferred plasmids by conjugation with a transfer rate (transconjugants per donor cells) ranging from 4.8 × 10^−1^ to 1.5 × 10^−9^ as observed from a human and environment isolate, respectively (Table 2).

### 2.3. Transferrable Resistance of Non-Beta-Lactam Phenotype among Isolates of Human, Animal and the Environment

Table 3 and Figure 1 presents a summary of non-beta lactam resistant phenotypes transferred along the *bla*_CTX-M-15_ gene. A total of 45 plasmids successful transferred the gene to transconjugants. Non-beta-lactam resistance phenotypes were observed in 34/45(75.6%) transconjugants. Donor resistance to ciprofloxacin (CIP), tetracycline (TE) and trimethoprim-sulphamethoxazole (SXT) was observed in 46/51 (90.2%), 47/51 (92.2%) and 48/51 (94.1%) of events, respectively, and was co-transferred in 29/34 (85.3%) of such events. Gentamicin was the least transferred with a frequency of 17/34 (50.0%).

### 2.4. Replicon Types of Plasmids Carrying bla_CTX-M-15_

Common replicon types were FIA (*n =* 11) and FIB (*n =* 27) that occurred as single transferrable replicons in 14 events. Inc A/C and Y replicons were minor, and each was typed once. Of all the 14 typed transconjugant plasmids, 7 replicons were observed in both donors and transconjugants, while 20/27 donor replicons were not observed in respective transconjugants (Table 4), (Figure 2).

### 2.5. Transfer Success of bla_CTX-M-15_ among Escherichia coli Isolates

Table 5 shows the percentage transfer of *bla*_CTX-M-15_ among *E. coli* donor isolates. Out of 42 *E. coli* donors, 37 (88.1%) successfully transferred the gene, accounting for an 82.1% of all transconjugants. All *E. coli* originating from animals transferred the gene successfully.

## 3. Discussion

In this study, we aimed to determine conjugation frequencies and type plasmids carrying the *bla*_CTX-M-15_ gene from human, animal and environment ESBL producing isolates. The study is epidemiologically important in understanding the pattern and possibly predict the flow of AMR from different sources. As presented in (Table 1), *Escherichia coli* was the dominant bacteria species from all sourced samples. The successful colonisation of *E. coli* in human and animal gastrointestinal tract (GIT) have been reported previously [25], the GIT can serve as exchange hotspots and reservoirs of antimicrobial resistance genes. Likewise, *Escherichia coli* and *Klebsiella pneumoniae* are frequently isolated in infections associated with CTX-M-15 in hospitals [26] and the community, including households [27], aquatic environment [28] and the soil [29]. 

The transfer efficiency of *bla*_CTX-M-15_ among isolates in this study was higher (88%) [Table 2], than that reported by Zurfluh and colleagues [30], where a 38.3% efficiency was observed, however, despite the varying frequencies of transfer, the reported efficiency is slightly lower than that previously reported (100%) for randomly selected hospital originating isolates [20]. It is reported that high conjugation rates above thresholds compensate fitness costs and establish a long-term persistence of plasmids in multiple hosts [13,31,32] through maintaining successive generations of bacteria with adaptive traits. Therefore, the high transfer efficiency is a fitness advantage that improves the persistence and dissemination potential of *bla*_CTX-M-15_ ESBL in human, animal and the environment interface. Moreover, transfer failure for some isolates‒ CN4, CN7, CN42, CN46, CN50 and CN51, could possibly be due to the gene’s integration in the chromosome [33] or transposition events that prevent plasmid mobility.

As summarised in Table 3 and Figure 1, transferable multidrug resistance phenotypes were also observed. The conjugative spread of *bla*_CTX-M-15_ gene by IncF plasmids along with tetracycline, aminoglycoside and quinolones have been reported [34]. These plasmids harbor several combinations of resistance determinants and transfer them to human, animals and environment isolates through the ecological interaction of bacteria in these settings. Moreover, the genetic environment of *bla*_CTX-M-15_ is dominated by multiple antibiotic resistance genes such as *aac (6′)-lb-cr, tet (A, B), qnrS, qnr* and *sul* genes [35,36,37], whose phenotypic expression denotes the existing selection pressure for these antibiotics. Such selection can increase their transfer rate and possibly account for the high co-transfer of non-beta lactam antibiotics observed in this study. 

We also observed single replicon IncF plasmids as common vectors of *bla*_CTXM-15_, (Table 4), (Figure 2). Replicon typing of plasmids carrying antimicrobial resistance genes is important for detecting, tracing or monitoring the spread of antimicrobial resistance. These observations are in line with findings in the same setting [20] and elsewhere [30,38] where multireplicon FIA and FIB plasmids were reported to carry multiple resistance genes including *bla*_CTX-M-15_. As in previous studies, IncY plasmids and Inc A/C carrying *bla*_CTX-M-15_ in association with quinolone and aminoglycoside genes were also reported [22], [39,40]. Together these findings support the diverse nature of plasmids adapted to spread and maintain *bla*_CTXM-15_ gene.

The absence of donor replicons in respective transconjugants might have resulted from conjugation failure, multi-replicons (undetected by the method used) that destabilise and prevent the transfer of some resident replicons [41], and prior plasmid dependent mutations (which do not occur in transconjugants) that may have altered backbone genes of donor plasmids and obscure the detection of existing replicons [32,39,41]. In addition, and as a shortcoming, the PBRT technique used in detecting plasmid replicons can give false-negative results when replicon sequences go undetected by the primer sets used, target replicon sequences undergo mutation through transpositional alterations and the unknown existence of new replicons in such plasmid [42].

Lastly, we observed that all animal originating *E. coli* isolates transferred *bla*_CTX-M-15_ to respective transconjugants (Table 5), (Figure 3). These findings were also supported by a recent review [34], that human and animal originating *E.coli* are adapted to disseminate ESBL genes by IncF plasmids. The colonisation and infection of animals by *E. coli* maximises microbial interactions between non-pathogenic and pathogenic commensal *E.coli* in either companion or food-producing animals and facilitate the exchange of materials between them through conjugation. In addition, the increasing use of antibiotics in animals could select and transfer resistant pathogenic bacteria from animals to human and the environment with huge cost implications. Since AMR is a public health threat, the highest transfer rate observed in animal originating *E.* coli calls for integrated efforts to address AMR with experts from veterinary, human and ecological fields. It further implies that animals may serve as dual targets for studies focusing on the horizontal transfer and evolution of antimicrobial resistance.

## 4. Materials and Methods

### 4.1. Study Isolates 

All isolates used as donors in this study were obtained from the Catholic University of Health and Allied Sciences (CUHAS) in Mwanza Tanzania. A total of 51 *bla*_CTX-M-15_ positive isolates were purposively selected and activated overnight in Luria Bertani (LB) broth at 37 °C ready for use in conjugation and PBRT techniques. 

Among the 51 isolates, twenty-two *bla*_CTX-M-15_ positive isolates were obtained from a study that reported the magnitude of fecal carriage and diversity of ESBL genotypes among human residing in rural communities of Mwanza Tanzania [19], 12 other *bla*_CTX-M-15_ positive isolates were from a study that reported the fecal carriage of ESBL among companion and domestic farm animals that included pigs, chicken, dogs and goats [21]. The remaining 17 environmental isolates were obtained from a study that investigated the presence of *bla*_CTX-M-15_ from muddy soils and gut contents of freshwater fish from Lake Victoria in Mwanza Tanzania [22].

### 4.2. Antibiotic Susceptibility Testing

Susceptibility testing of all donor isolates and the resulting transconjugants was performed by the disk diffusion method on Mueller Hinton agar as recommended by the Clinical and Laboratory Standard Institute [43]. Antibiotics tested were tetracycline (30 µg), gentamicin (30 µg), ciprofloxacin (5 µg) and trimethoprim/sulphamethoxazole (1.25/23.75 μg) (Hi-media, India).

### 4.3. Conjugation Experiment

A total of 51 known *bla_CTX-M-15_* positive isolates and *Escherichia coli* J53 ((F^−^, *met*, *pro*, Az^r^)‒a mutant strain of *E. coli* [44] obtained from the Institute of Medical Microbiology, Giessen, Germany, were used as donors and recipient strain, respectively. As previously described [20], we performed conjugation experiments with some modifications. Shortly, the recipient strain was prepared by streaking *Escherichia coli* J53 in Luria Bertani (LB) plates supplemented with 100 µg/mL NaN_3_ (LB++), while donor strains were selected in LB plates supplemented with 2 µg/mL cefotaxime only (LB+). From these, fresh overnight donor and recipient strains were prepared by picking single colonies emulsified in 10 mL LB broth and incubated overnight at 37 °C in a 150 rpm shaking incubator. After exactly 12 h, equal volumes (500 µL) of donor and recipient strains were immediately mixed in 1.5 mL eppendorf tubes previously labeled transconjugant (Tc) while 1000 µL of donor strain were added in fresh tubes of similar volume‒to be separately selected on LB+ and LB++ plates as respective controls. All tubes were incubated at 37 °C for 15 min, vortexed briefly, centrifuged at 12,000 g for 2 min and the pellet re-suspended in fresh 1000 µL LB broth. Finally, 0.1 mL of 10^−1^ to 10^−4^ transconjugant cultures were double selected on LB plates supplemented with 100 µg/mL NaN_3_ and 2 µg/mL cefotaxime. Conjugation efficiency was reported as transconjugants per donor cells, with the denominator obtained from an initial volume of 100 µL.

### 4.4. Genomic Extraction of Donor and Transconjugants DNA 

Donor and transconjugant genomic DNA was extracted using a previously described chelex protocol with slight modifications [45]. First, 5 µL of proteinase K (10 mg/mL) were added into tubes containing 100 µL fresh LB emulsified colonies. In the same tubes, 300 µL of chelex buffer (Qiagen GmbH, Hilden, Germany) was added consecutively. The mixture was incubated for 3 hr at 55 °C before adding 85 µL of 5 M NaCl and vortexed for 15 s to precipitate proteins. The supernatant was centrifuged at 13,000× *g* for 10 min followed by the addition of 300 µL of 100% cold ethanol and a 5 min centrifugation at 13,000× *g* that precipitated and pelleted the DNA. Lastly, the pellet was rinsed by pouring off the remaining fluid, adding 500 µL of 70% ethanol, centrifuging at 13,000× *g* for 5 min and leaving the pellet to air dry at 55 °C for 10 min. The DNA was then re-suspended in 50 µL nuclease-free water. Nanodrop (Thermo Scientific, Wilmington, DE) was used to check the quantity of the DNA, while the quality was confirmed by electrophoresis in 1.5% (*w*/*v*) agarose gel using TAE buffer. The obtained DNA samples were used in typing plasmid replicons or stored at −20 °C.

### 4.5. PCR Based Replicon Typing

Targeted genes were amplified by a simplified version of the previously described PBRT technique [42]. Shortly, the eight Polymerase Chain Reaction (PCR) panels illustrated by Caratolli and colleagues [46], were reduced to three [42], (Table 6). PCR was performed using a readily reconstituted master mix according to manufacturer’s instructions (New England BioLabs, Inc. Beverly, MA) under the following conditions; 5 min at 94 °C; 30 cycles of 30 s at 94 °C, 30 s at 60 °C and 90 s at 72 °C; then a final extension of 5 min at 72 °C. Amplicons were visualised on 1.5% tris-acetate EDTA agarose gels alongside a 100 bp DNA ladder (New England BioLabs, Inc. Beverly, MA). The sample was considered positive for replicon gene (s) if an amplicon of the expected band size was observed.

## 5. Conclusions

Majority of plasmids carrying *bla*_CTX-M-15_ were conjugatively transferred by IncF plasmids along with non-beta lactam resistance. The heterogeneous nature of these plasmids continuously maintains and reserves the *bla*_CTX-M-15_ gene in these settings. The 100% transfer efficiency among *E. coli* of animal origin is of concern since the networked interaction of animals with human and their environment continuously exchange and reserve resistance determinants in this interface. Therefore, there is a need for more research to understand the interaction and spread of mobile elements circulating in animals, One Health approach is to be intensified to further address AMR as a public health threat.

## Figures and Tables

**Figure 1 antibiotics-10-00836-f001:**
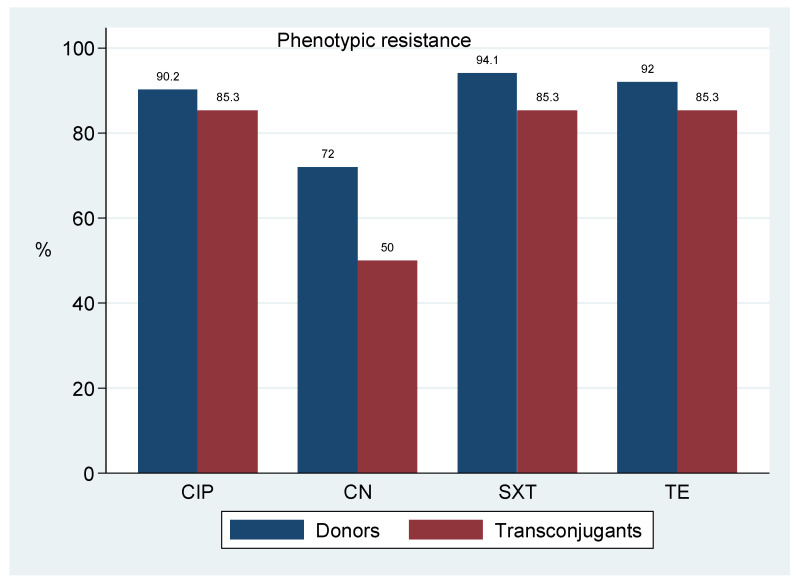
Transfer success of *bla*_CTX-M-15_ among *E. coli* isolates of human, animals and the environment.

**Figure 2 antibiotics-10-00836-f002:**
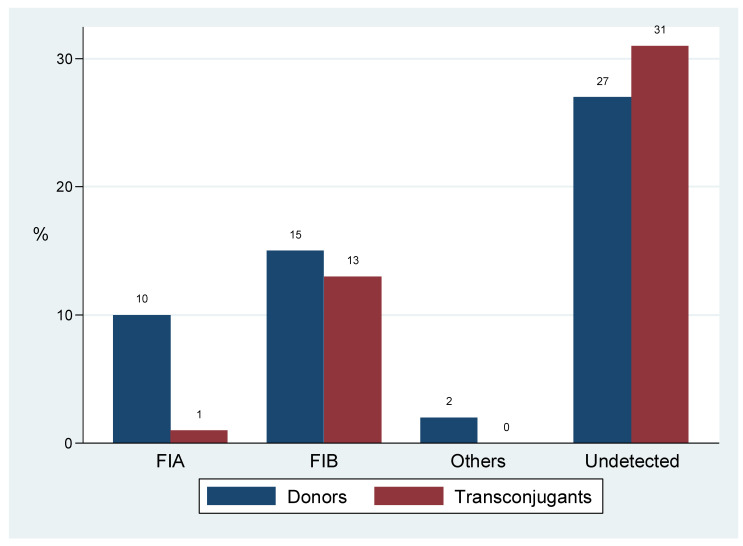
Replicon types of plasmids carrying *bla*_CTX-M-15_ among donors and transconjugants.

**Figure 3 antibiotics-10-00836-f003:**
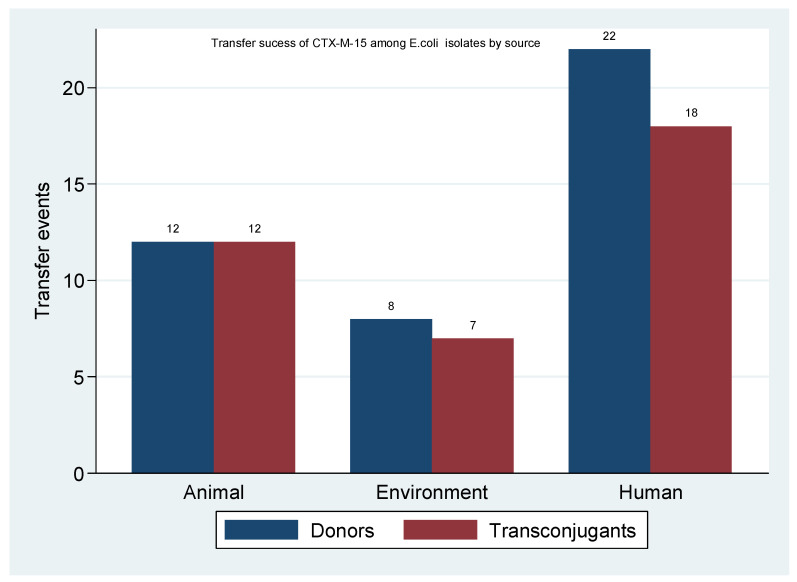
Transfer success of *bla*_CTX-M-15_ among *E. coli* isolates of human, animals and the environment.

**Table 1 antibiotics-10-00836-t001:** Bacteria species distributed among donor isolates of human, animal and environment.

Sample Origin	Sample Type	Frequency n%	Species	Species n (%)	Total n (%)
Human	Human	22 (43.14)	*E. coli*	22 (43.1)	22 (43.14)
	Goat	1 (1.96)	*E. coli*	1 (1.96)	
Animal	Pig	3 (5.88)	*E. coli*	3 (5.88)	
	Dog	6 (11.76)	*E. coli*	6 (11.76)	12 (23.52)
	Chicken	2 (3.92)	*E. coli*	2 (3.92)	
Environment	Soil	6 (11.76)	*E. coli*	6 (11.76)	
			*E. coli*	2 (3.92)	
	Fish	11 (21.57)	*K. pneumoniae*	3 (5.88)	17 (33.32)
			*C. braakii*	2 (3.92)	
			*E. cloacae*	4 (7.84)	
**Total (n)**		**51 (100)**			**51 (100)**

**Table 2 antibiotics-10-00836-t002:** Conjugation efficiency of human, animal and environment donor isolates.

Sample ID	Source	Species	Conjugation Frequency
CN1	Fish	*E. cloacae*	8.2 × 10^−5^
CN2	Fish	*E. cloacae*	2.3 × 10^−4^
CN3	Fish	*E. cloacae*	5.2 × 10^−5^
CN4	Fish	*E. cloacae*	NIL
CN5	Fish	*C. braakii*	7.5 × 10^−6^
CN6	Fish	*E. coli*	7.6 × 10^−3^
CN7	Fish	*E. coli*	NIL
CN8	Fish	*K. pneumoniae*	2.0 × 10^−5^
CN9	Fish	*K. pneumoniae*	4.2 × 10^−4^
CN10	Fish	*K. pneumoniae*	3.3 × 10^−5^
CN11	Fish	*C. braakii*	9.4 × 10^−4^
CN12	Pig	*E. coli*	4.7 × 10^−5^
CN13	Pig	*E. coli*	2.6 × 10^−6^
CN14	Pig	*E. coli*	9.8 × 10^−5^
CN15	Local chicken	*E. coli*	4.7 × 10^−5^
CN16	Local chicken	*E. coli*	8.4 × 10^−7^
CN17	Goat	*E. coli*	4.1 × 10^−6^
CN18	Dog	*E. coli*	2.1 × 10^−5^
CN19	Dog	*E. coli*	1.2 × 10^−7^
CN20	Dog	*E. coli*	5.0 × 10^−5^
CN21	Dog	*E. coli*	1.1 × 10^−6^
CN22	Dog	*E. coli*	6.0 × 10^−4^
CN23	Dog	*E. coli*	9.6 × 10^−6^
CN24	Environment	*E. coli*	1.5 × 10^−9^
CN25	Environment	*E. coli*	2.6 × 10^−7^
CN26	Environment	*E. coli*	3.5 × 10^−6^
CN27	Environment	*E. coli*	2.9 × 10^−7^
CN28	Environment	*E. coli*	6.1 × 10^−6^
CN29	Environment	*E. coli*	7.2 × 10^−3^
CN30	Human	*E. coli*	1.0 × 10^−3^
CN31	Human	*E. coli*	4.7 × 10^−4^
CN32	Human	*E. coli*	2.1 × 10^−4^
CN33	Human	*E. coli*	4.0 × 10^−5^
CN34	Human	*E. coli*	5.4 × 10^−5^
CN35	Human	*E. coli*	4.8 × 10^−1^
CN36	Human	*E. coli*	1.7 × 10^−4^
CN37	Human	*E. coli*	3.5 × 10^−7^
CN38	Human	*E. coli*	8.1 × 10^−5^
CN39	Human	*E. coli*	1.2 × 10^−5^
CN40	Human	*E. coli*	2.7 × 10^−5^
CN41	Human	*E. coli*	2.4 × 10^−7^
CN42	Human	*E. coli*	NIL
CN43	Human	*E. coli*	5.5 × 10^−6^
CN44	Human	*E. coli*	4.4 × 10^−6^
CN45	Human	*E. coli*	2.9 × 10^−6^
CN46	Human	*E. coli*	NIL
CN47	Human	*E. coli*	2.1 × 10^−5^
CN48	Human	*E. coli*	1.2 × 10^−4^
CN49	Human	*E. coli*	1.1 × 10^−7^
CN50	Human	*E. coli*	NIL
CN51	Human	*E. coli*	NIL

NIL: no conjugation.

**Table 3 antibiotics-10-00836-t003:** Antibiotic resistance phenotypes of donors and transconjugants of human animals and the environment.

Sample No.	Source	Species	Donor’s Non-B-lactam Resistance Phenotype
CN1	Fish	*E. cloacae*	SXT *, CIP *,CN *,TE *
CN2	Fish	*E. cloacae*	CIP, SXT, CN, TE
CN3	Fish	*E. cloacae*	CIP *, SXT *, TE *, CN*
CN4	Fish	*E. cloacae*	CIP, CN, TE, SXT
CN5	Fish	*C. braakii*	CIP *, SXT *, CN *, TE *
CN6	Fish	*E. coli*	CIP, SXT, CN, TE
CN7	Fish	*E. coli*	CIP, TE
CN8	Fish	*K. pneumoniae*	CIP *, SXT *, CN *, TE *
CN9	Fish	*K. pneumoniae*	CIP *, SXT *, CN *, TE *
CN10	Fish	*K. pneumoniae*	CIP, SXT, CN, TE
CN11	Fish	*C. braakii*	CIP, SXT, CN, TE *
CN12	Pig	*E. coli*	CIP *^,^ SXT *, TE *
CN13	Pig	*E. coli*	TE, CIP, CN
CN14	Pig	*E. coli*	CIP *, SXT *, TE *, CN *
CN15	Local chicken	*E. coli*	CIP, SXT, CN, TE
CN16	Local chicken	*E. coli*	CIP, SXT, CN, TE
CN17	Goat	*E. coli*	SXT, TE *, CN, CIP *
CN18	Dog	*E. coli*	SXT
CN19	Dog	*E. coli*	SXT *, CIP *, TE, CN
CN20	Dog	*E. coli*	CIP *, SXT *, TE *
CN21	Dog	*E. coli*	CIP *, SXT *, TE *, CN *
CN22	Dog	*E. coli*	CIP *, CN *, TE *, SXT *
CN23	Dog	*E. coli*	SXT, TE, CN, CIP
CN24	Environment	*E. coli*	SXT *, CIP *, TE *
CN25	Environment	*E. coli*	SXT, TE, CIP
CN26	Environment	*E. coli*	CIP *, SXT *, TE*
CN27	Environment	*E. coli*	CIP *
CN28	Environment	*E. coli*	CIP *, SXT *, CN *, TE *
CN29	Environment	*E. coli*	CN, CIP *, SXT *, TE *
CN30	Human	*E. coli*	TE *, CIP *, CN, SXT *
CN31	Human	*E. coli*	CIP *, SXT *
CN32	Human	*E. coli*	SXT *, CIP *
CN33	Human	*E. coli*	TE *, CN *, CIP *, SXT *
CN34	Human	*E. coli*	SXT *, TE *, CN *, CIP
CN35	Human	*E. coli*	CIP *, CN *, SXT *, TE *
CN36	Human	*E. coli*	CIP *, CN *, SXT *, TE *
CN37	Human	*E. coli*	CIP *, CN *, SXT *, TE *
CN38	Human	*E. coli*	SXT *, TE *, CIP*, CN *
CN39	Human	*E. coli*	SXT, TE, CIP *, CN *
CN40	Human	*E. coli*	SXT *, TE *
CN41	Human	*E. coli*	SXT, TE *, CIP *, CN
CN42	Human	*E. coli*	SXT, CIP, CN, TE
CN43	Human	*E. coli*	CN *, CIP *, SXT *, TE *
CN44	Human	*E. coli*	SXT, TE, CIP
CN45	Human	*E. coli*	SXT, TE, CIP, CN
CN46	Human	*E. coli*	TE, SXT
CN47	Human	*E. coli*	SXT *, TE *, CIP, CN
CN48	Human	*E. coli*	SXT *, TE *, CIP, CN
CN49	Human	*E. coli*	CIP *, CN *, SXT *, TE *
CN50	Human	*E. coli*	SXT, TE
CN51	Human	*E. coli*	CN, CIP, SXT, TE

* Transferable resistance; SXT: Trimethoprim-sulphamethoxazole, CIP: Ciprofloxacin, TE: tetracycline, CN: Gentamicin.

**Table 4 antibiotics-10-00836-t004:** Replicon types of plasmids carrying *bla*_CTX-M-15_ among donors and transconjugants.

Sample Source	Conjugation Efficiency	Conjugation Range	Donor’s Plasmid Replicon	Transconjugant Replicon Type
Human	1.2 × 10^−4^		FIB	FIA
Human	8.1 × 10^−5^		FIA, FIB	FIB
Dog	5.0 × 10^−5^		FIB	FIB
Human	5.4 × 10^−5^	10^−6^–10^−3^	FIB	FIB
Human	2.1 × 10^−4^		FIB	FIB
Environment	7.2 × 10^−3^		FIB	FIB
Dog	1.1 × 10^−6^		FIB	FIB
Human	1.7 × 10^−4^		FIB	FIB
Dog	9.6 × 10^−6^		no rep	FIB
Dog	2.1 × 10^−5^		no rep	FIB
Human	1.2 × 10^−5^	10^−7^–10^−4^	no rep	FIB
Human	4.7 × 10^−4^		no rep	FIB
Environment	2.9 × 10^−7^		no rep	FIB
Fish	2.3 × 10^−4^		no rep	FIB
Fish	NIL		FIA, Y	NA
Human	NIL		no rep	NA
Human	NIL	0	no rep	NA
Human	NIL		no rep	NA
Human	NIL		no rep	NA
Fish	NIL		no rep	NA
Fish	4.2 × 10^−4^		A/C, FIA	no rep
Pig	2.6 × 10^−6^		FIA	no rep
Human	5.5 × 10^−6^		FIA	no rep
Dog	6.0 × 10^−4^		FIA	no rep
Pig	9.8 × 10^−5^		FIA	no rep
Human	2.9 × 10^−6^		FIA	no rep
Human	4.0 × 10^−5^		FIA	no rep
Human	4.8 × 10^−1^	10^−9^–10^−1^	FIA	no rep
Dog	1.2 × 10^−7^		FIB	no rep
Human	3.5 × 10^−7^		FIB	no rep
Environment	1.5 × 10^−9^		FIB	no rep
Environment	2.6 × 10^−7^		FIB	no rep
Human	4.4 × 10^−6^		FIB	no rep
Environment	3.5 × 10^−6^		FIB	no rep
Human	2.1 × 10^−5^		FIB	no rep
Fish	7.5 × 10^−6^		no rep	no rep
Fish	9.4 × 10^−4^		no rep	no rep
Human	2.7 × 10^−5^		no rep	no rep
Local chicken	4.7 × 10^−5^		no rep	no rep
Pig	4.7 × 10^−5^		no rep	no rep
Human	2.4 × 10^−7^		no rep	no rep
Fish	3.3 × 10^−5^		no rep	no rep
Fish	2.0 × 10^−5^	10^−7^–10^−3^	no rep	no rep
Fish	7.6 × 10^−3^		no rep	no rep
Human	1.1 × 10^−7^		no rep	no rep
Fish	5.2 × 10^−5^		no rep	no rep
Goat	4.1 × 10^−6^		no rep	no rep
Environment	6.1 × 10^−6^		no rep	no rep
Local chicken	8.4 × 10^−7^		no rep	no rep
Human	1.0 × 10^−3^		no rep	no rep
Fish	8.2 × 10^−5^		no rep	no rep

NIL: no conjugation, NA: no transconjugants, no rep: no typable replicon.

**Table 5 antibiotics-10-00836-t005:** Transfer success of *bla*_CTX-M-15_ among *E. coli* isolates of human, animals and the environment.

Source	*E. coli* Donors n	*E. coli* Transconjugants n (%)
Human	22	18 (81.8)
Animal	12	12 (100.0)
Environment	8	7 (87.5)
Total	42	37(88.1)

**Table 6 antibiotics-10-00836-t006:** Primers used in PCR based replicon typing of donor and transconjugant plasmids.

Primer Panel/Target	Direction	Primer Sequence	Annealing Temp (°C)	Amplicon Size (bp)
**Panel 1**				
B/O	F	5′-gcggtccggaaagccagaaaac-3′	60	159
	R	5′-tctgcgttccgccaagttcga-3′		
FIC	F	5′-gtgaactggcagatgaggaagg-3′	60	262
	R	5′-ttctcctcgtcgccaaactagat-3′		
A/C	F	5′-gagaaccaaagacaaagacctgga3′	60	465
	R	5′-acgacaaacctgaattgcctcctt-3′		
P	F	5′ctatggccctgcaaacgcgccagaaa3′	60	534
	R	5′-tcacgcgccagggcgcagcc-3′		
T	F	5′-ttggcctgtttgtgcctaaaccat-3′	60	750
	R	5′-cgttgattacacttagctttggac-3′		
**Panel 2**				
K/B	F	5′-gcggtccggaaagccagaaaac-3′	60	160
	R	5′-tctttcacgagcccgccaaa-3		
W	F	5′-cctaagaacaacaaagcccccg-3′	60	242
	R	5′-ggtgcgcggcatagaaccgt-3′		
FII_S_	F	5′-ctgtcgtaagctgatggc-3′	60	270
	R	5′-ctctgccacaaacttcagc-3′		
FIA	F	5′-ccatgctggttctagagaaggtg-3′	60	462
	R	5′-gtatatccttactggcttccgcag-3′		
FIB	F	5′-ggagttctgacacacgattttctg-3′	60	702
		5′-ctcccgtcgcttcagggcatt-3′		
Y	F	5′-aattcaaacaacactgtgcagcctg-3′	60	765
	R	5′-gcgagaatggacgattacaaaacttt-3′		
**Panel 3**				
I1	F	5′-cgaaagccggacggcagaa-3′	60	139
	R	5′-tcgtcgttccgccaagttcgt-3′		
F_repB_	F	5′-tgatcgtttaaggaattttg-3′	60	270
	R	5′-gaagatcagtcacaccatcc-3′		
X	F	5′-aaccttagaggctatttaagttgctgat-3′	60	376
	R	5′-tgagagtcaatttttatctcatgttttagc3′		
HI1	F	5′-ggagcgatggattacttcagtac-3′	60	471
	R	5′-tgccgtttcacctcgtgagta-3′		
N	F	5′-gtctaacgagcttaccgaag-3′	60	559
	R	5′-gtttcaactctgccaagttc-3′		
HI2	F	5′-tttctcctgagtcacctgttaacac-3′	60	644
	R	5′-ggctcactaccgttgtcatcct-3′		
L/M	F	5′-ggatgaaaactatcagcatctgaag-3′	60	785
	R	5′-ctgcaggggcgattctttagg-3′		

## Data Availability

All data supporting presented results is available in this article.

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
