# Peer review of "Conjugative Plasmids Disseminating CTX-M-15 among Human, Animals and the Environment in Mwanza Tanzania: A Need to Intensify One Health Approach"

_antibiotics, 2021, doi:10.3390/antibiotics10070836_

Round 1

Reviewer 1 Report

Review of the article: “CONJUGATIVE PLASMIDS DISSEMINATING CTX-M-15 AMONG HUMAN, ANIMALS AND THE ENVIRONMENT IN MWANZA TANZANIA: A NEED TO INTENSIFY ONE HEALTH APPROACH”

Submission ID - antibiotics-1227174

The manuscript is interesting, well written and concerns a very important topic – disseminating of CTX-M-15. The experiments were well planned and performed. Below I have presented some issues which could be taken into account preparing the final version of the manuscript.

Abstract

All abbreviations presented in the Abstract must be explained including ESBL

Line 22 – “The conjugation efficiency of plasmids carrying blaCTX-M-15 was 88.2% (54/51)” – something is wrong in this calculation. I have checked the results it was 45/51

Line 23 – “of 10‒1 to 10‒9 and” – this form of presentation is not clear for potential readers, please use italic for names of bacteria – E.coli

Lines 24 – 25 – “Inc FIA or Inc FIB” should be explained

Line 30 – “One Health approach” – not clear, please explain (in other parts than Abstract of course)

Introduction

Line 43 - beta-lactamase-producing soil organisms ?

Lines 57-58 – „Plasmids are responsible for the intracellular accumulation and intercellular transfer of antimicrobial resistance (AMR) by the process of conjugation (Vrancianu et al., 2020).” – would it be possible to rephrase this sentence/write it more clearly – only suggestion

General comment – the authors concentrated on situation in Tanzania – e.g. Fig 1. Would it be possible to provide some more general information?

“One Health approach” – please provide some more details

Materials and methods

Section 4.2 – please provide information about the source of the discs (producer)

Results

I have some doubts about “construction” of Table 1. In first column the authors presented “Sample origin” and three different origins are presented. I do not understand why fish is not included to other animals. I think that only soil should be included to “environment”, all animals should be included to one group.  I have also one (minor of importance) “graphical” comment – in current version goat is closer to human than other animals.

Table 2 – CN42 – there is Nil in other cases you used NIL

Table 4 -  “no rep” should be defined

Line 109 – “served in 34/45 transconjugants” – the percentage could be presented  

Discussion

Generally the discussion is interesting. However, in this section the authors should more clearly present their opinions (some sentences are not clear for me), e.g.: “Therefore, it is possible that the food chain as previously explained by Irrgang et al. (2017) is the reservoir of blaCTX-M-15 gene in animals passing it to human and the environment. – but the gene is not transferred between animals and humans; or “these species can spread quickly between commensal and pathogenic bacteria and their environment”

Conclusions

Conclusions are well presented – no critical remarks

Final decision – minor revision.

Author Response

Review of the article: “CONJUGATIVE PLASMIDS DISSEMINATING CTX-M-15 AMONG HUMAN, ANIMALS AND THE ENVIRONMENT IN MWANZA TANZANIA: A NEED TO INTENSIFY ONE HEALTH APPROACH”

Submission ID - antibiotics-1227174

The manuscript is interesting, well written and concerns a very important topic – disseminating of CTX-M-15. The experiments were well planned and performed. Below I have presented some issues which could be taken into account preparing the final version of the manuscript.

Abstract

All abbreviations presented in the Abstract must be explained including ESBL

RESPONSE: All abbreviations in the abstract have been removed

Line 22 – “The conjugation efficiency of plasmids carrying blaCTX-M-15 was 88.2% (54/51)” – something is wrong in this calculation. I have checked the results it was 45/51

RESPONSE: This is now corrected as highlighted

Line 23 – “of 10‒1 to 10‒9 and” – this form of presentation is not clear for potential readers, please use italic for names of bacteria – E.coli

RESPONSE: The presented form i.e.  10‒1 to 10‒9 is the range of conjugation frequency (transconjugant/donor cell) observed from environment and human isolate respectively. We therefore consider it correct as it is unless there is a more clear suggestion for us to willingly take.

Lines 24 – 25 – “Inc FIA or Inc FIB” should be explained

RESPONSE: We have corrected this as highlighted in the abstract. We hope the sentence is now clear

Line 30 – “One Health approach” – not clear, please explain (in other parts than Abstract of course)

RESPONSE: The concept is explained in the introduction, discussion and finally at the conclusion

Introduction

Line 43 - beta-lactamase-producing soil organisms?

RESPONSE: Yes, the soil resistome consists of multiple resistant determinants especially beta-lactamases. In natural (non-selective environments) these occurs as a result of minute concentrations of compounds that select for resistance. In the soil environment, a rich source of microorganisms producing beta lactam antibiotics like penicillin and cephalosporins is reported as (new references added to support this). It is in the soil environment as well where plasmid mediated beta-lactamases evolved. This indicates an active significant presence of bacteria producing these enzymes.

Lines 57-58 – Plasmids are responsible for the intracellular accumulation and intercellular transfer of antimicrobial resistance (AMR) by the process of conjugation (Vrancianu et al., 2020).” – would it be possible to rephrase this sentence/write it more clearly – only suggestion

RESPONSE: The sentence is rephrased more clearly as highlighted

General comment – the authors concentrated on situation in Tanzania – e.g. Fig 1. Would it be possible to provide some more general information?

RESPONSE: We have added new information in the introduction

“One Health approach” – please provide some more details

RESPONSE:  We have tried elaborating the concept in the last paragraph of the introduction as a justification to the study findings

Materials and methods

Section 4.2 – please provide information about the source of the discs (producer)

RESPONSE: The source have been provided

Results

I have some doubts about “construction” of Table 1. In first column the authors presented “Sample origin” and three different origins are presented. I do not understand why fish is not included to other animals. I think that only soil should be included to “environment”, all animals should be included to one group.  I have also one (minor of importance) “graphical” comment – in current version goat is closer to human than other animals.

RESPONSE: 1.for the graphical comment, the demarcation is clear from the table

  1. Fish is included in the environment originating sample to represent aquatic environment where it resides, effluents and surface run off drains into the lake and possibly the net microbial composition of aquatic organisms (animals and plants) is shaped by this environment.

Table 2 – CN42 – there is Nil in other cases you used NIL

RESPONSE: Nil is now replaced with NIL as highlighted

Table 4 - “no rep” should be defined

RESPONSE: Defined “no rep” as highlighted

Line 109 – “served in 34/45 transconjugants” – the percentage could be presented  

RESPONSE: The percent is added as highlighted

Discussion

Generally the discussion is interesting. However, in this section the authors should more clearly present their opinions (some sentences are not clear for me), e.g.: “Therefore, it is possible that the food chain as previously explained by Irrgang et al. (2017) is the reservoir of blaCTX-M-15 gene in animals passing it to human and the environment. – but the gene is not transferred between animals and humans; or “these species can spread quickly between commensal and pathogenic bacteria and their environment”

RESPONSE: We have improved the first part of the discussion (page 7 and 8) as highlighted and hope it is now clear

Conclusions

Conclusions are well presented – no critical remarks

RESPONSE: We thank the reviewer for his/her positive input on this section

Final decision – minor revision.

Reviewer 2 Report

This manuscript is great as is. They ask relevant questions and addresses an area of great concern. There are many avenues to follow up and I look forward to seeing more of their work investigating antibiotic resistance in the area. The English needs to be cleaned up a bit throughout the manuscript, but overall it's a great study with legitimate design and reasonable conclusions. 

Author Response

Review of the article: “CONJUGATIVE PLASMIDS DISSEMINATING CTX-M-15 AMONG HUMAN, ANIMALS AND THE ENVIRONMENT IN MWANZA TANZANIA: A NEED TO INTENSIFY ONE HEALTH APPROACH”

Submission ID - antibiotics-1227174

This manuscript is great as it is. They ask relevant questions and addresses an area of great concern. There are many avenues to follow up and I look forward to seeing more of their work investigating antibiotic resistance in the area. The English needs to be cleaned up a bit throughout the manuscript, but overall it's a great study with legitimate design and reasonable conclusions. 

RESPONSE: We thank the reviewer for his/her positive remark on this manuscript overall. We have checked the grammar and other language requirements throughout the manuscript as instructed. We  look forward to work more on this area.

Reviewer 3 Report

The article by Minja et al. describes the isolation of E.coli from multiple sources in Tanzania and describes the antibiotic susceptibility testing and plasmid conjugation levels in the isolates. Overall, the conjugation frequencies seem high. This article is important locally and globally to understand the dramatic distribution and extent of antibiotic resistance.

  1. Add graphs to visualize the data in tables 2, 3 and 4.
  2. The authors have provided a snap shot of antibiotic resistance coding plasmids and have done antibiograms from different hosts/environments. Explain in a sentence or two of the discussion how this study contributes to “interventions” (mentioned in line 87) that are One-Health based. This is an epidemiological study and is not really about interventions (that is, the antibiotics prescribed to hosts is not a factor that is accounted for).

The paper needs significant grammar corrections and changes to sentence syntax to be readable by a general global audience. There are some sentences (such as in line 17) that are hard to understand. A few corrections are highlighted below but the authors are encouraged to double check grammar throughout the paper using standard British or American grammar rules so that a global audience can understand it as well.

Line 13: What is “mobile genetic elements several niches.”? Two sentences seem to have been combined here.

Line 17: What is “ positive achieved isolates”?

Line 27: Check grammar on “Gentamicin was transferred”. The antibiotic was transferred?

Line 29, 30 : These are two different sentences. They need a period in between, not a comma.

Line 35 and throughout: Capitalize E in bacterial family name

Line 93 – 95: Starting at “whereas”, the sentence makes little sense. Check the grammar.

Line 122: “Gentamicin” not “gentamycin”.

Line 204, 210, 215: Cities/towns/Country information needed.

Line 227: “Streaking” not “striking”

Author Response

Review of the article: “CONJUGATIVE PLASMIDS DISSEMINATING CTX-M-15 AMONG HUMAN, ANIMALS AND THE ENVIRONMENT IN MWANZA TANZANIA: A NEED TO INTENSIFY ONE HEALTH APPROACH”

Submission ID - antibiotics-1227174

The article by Minja et al. describes the isolation of E.coli from multiple sources in Tanzania and describes the antibiotic susceptibility testing and plasmid conjugation levels in the isolates. Overall, the conjugation frequencies seem high. This article is important locally and globally to understand the dramatic distribution and extent of antibiotic resistance.

  1. Add graphs to visualize the data in tables 2, 3 and 4.

RESPONSE: Graphs have been added as instructed

  1. The authors have provided a snap shot of antibiotic resistance coding plasmids and have done antibiograms from different hosts/environments. Explain in a sentence or two of the discussion how this study contributes to “interventions” (mentioned in line 87) that are One-Health based. This is an epidemiological study and is not really about interventions (that is, the antibiotics prescribed to hosts is not a factor that is accounted for).

RESPONSE: The importance of the study is highlighted in the first three sentences of the discussion as suggested

The paper needs significant grammar corrections and changes to sentence syntax to be readable by a general global audience. There are some sentences (such as in line 17) that are hard to understand. A few corrections are highlighted below but the authors are encouraged to double check grammar throughout the paper using standard British or American grammar rules so that a global audience can understand it as well.

RESPONSE: The manuscript have been checked for grammar and other mistakes

Line 13: What is “mobile genetic elements several niches.”? Two sentences seem to have been combined here.

RESPONSE: The sentence have been corrected

Line 17: What is “positive achieved isolates”?

RESPONSE: The sentence is corrected as highlighted

Line 27: Check grammar on “Gentamicin was transferred”. The antibiotic was transferred?

RESPONSE: The grammar is corrected

Line 29, 30: These are two different sentences. They need a period in between, not a comma.

RESPONSE: The sentences are now separated with a period

Line 35 and throughout: Capitalize E in bacterial family name

RESPONSE: This is checked and capitalized as required

Line 93 – 95: Starting at “whereas”, the sentence makes little sense. Check the grammar.

RESPONSE: The sentence is checked and corrected

Line 122: “Gentamicin” not “gentamycin”.

RESPONSE: The error is corrected throughout the manuscript

Line 204, 210, 215: Cities/towns/Country information needed.

RESPONSE: City and country information have been added

Line 227: “Streaking” not “striking”

RESPONSE: The correct word is used

Reviewer 4 Report

June 10, 2021

Journal: Antibiotics

Title: Conjugative plasmids disseminating ctx-m-15 among human, animals and the environment in mwanza tanzania: a need to intensify one health approach

Authors: Caroline A. Minja, Gabriel Shirima and Stephen E. Mshana

Dear Editor,

The current manuscript explores the dissemination of resistance genes by determining the conjugation frequencies and replicon types associated with plasmids carrying blaCTX-M-15 gene from ESBL isolates. The manuscript carries the scientific merit and would be suitable for publication after considering the comments suggested below.

Comments to authors:

  1. What does the abbreviation ESBL in the abstract mean?
  2. English editing is highly recommended
  3. The keywords need to be more specific for example “One-health” is very general
  4. Author guidelines should be revised and applied carefully, the abstract for example should be structured.
  5. The manuscript is full of very long and complex sentences which is barely understandable
  6. The experimental part should be accompanied with more references
  7. What type of nanodrop was used in the study?
  8. The introduction should be comprehensible to scientists working outside the topic of the paper. Also, the importance of the study should be highlighted in the introduction and its impact on public health should be emphasized as well
  9. This reference would be of benefit:

Zajmi, A., Hashim, N.M., Noordin, M., Khalifa, S.A., Ramli, F., Ali, H.M., El-Seedi, H.R. (2015): Ultrastructural study on the antibacterial activity of Artonin E versus streptomycin against Staphylococcus aureus strains. Plos One 10, e0128157

Author Response

Dear Editor,

The current manuscript explores the dissemination of resistance genes by determining the conjugation frequencies and replicon types associated with plasmids carrying blaCTX-M-15 gene from ESBL isolates. The manuscript carries the scientific merit and would be suitable for publication after considering the comments suggested below.

Comments to authors:

  1. What does the abbreviation ESBL in the abstract mean?

RESPONSE: The abbreviation is explained and not used in the abstract

2. English editing is highly recommended

RESPONSE: English is edited throughout the manuscript as suggested

3.  The keywords need to be more specific for example “One-health” is very general

RESPONSE:  We have discussed and find that One Health is specific as it is, we request for suggestions of synonyms for this concept

4.  Author guidelines should be revised and applied carefully, the abstract for example should be structured.

RESPONSE: We have included a structured abstract as required

5. The manuscript is full of very long and complex sentences which is barely understandable

RESPONSE: We have double checked the manuscript for such sentences and corrected them

6. The experimental part should be accompanied with more references

RESPONSE: We performed the experiment using different techniques as cited. The conjugation and DNA extraction parts of the methodology were performed with MODIFICATIONS from the cited reference (s) as explained in these sections. This is possibly the reason for not citing many references as expected

7. What type of nanodrop was used in the study?

RESPONSE: We have included the required information

8. The introduction should be comprehensible to scientists working outside the topic of the paper. Also, the importance of the study should be highlighted in the introduction and its impact on public health should be emphasized as well

RESPONSE: We have re-written the introduction and included this information in the first and last paragraph of the introduction

9. This reference would be of benefit:

Zajmi, A., Hashim, N.M., Noordin, M., Khalifa, S.A., Ramli, F., Ali, H.M., El-Seedi, H.R. (2015): Ultrastructural study on the antibacterial activity of Artonin E versus streptomycin against Staphylococcus aureus strains. Plos One 10, e0128157

RESPONSE: Thank you for the reference, it was useful for understanding antimicrobial resistance and re-structuring the importance of this study in the introduction part

Reviewer 5 Report

I think the study could give important information to the scientific community. Current study lacking scientific standards in description and presentation in the discussion part. I think that manuscript could be improved by some changes listed below:

1-The manuscript must be revised completely before it can be submitted and reviewed again. I would suggest to re-write its introductions, discussion and conclusion.

2-The language must be improved drastically. Some of the sentences are misleading and often not to understand. It recommended an English language revision since frequently the use of the grammar and the syntax results quite weird.

Author Response

Review of the article “CONJUGATIVE PLASMIDS DISSEMINATING CTX-M-15 AMONG HUMAN, ANIMALS AND THE ENVIRONMENT IN MWANZA TANZANIA: A NEED TO INTENSIFY ONE HEALTH APPROACH”

Submission ID: Antibiotics-1227174

I think the study could give important information to the scientific community. Current study lacking scientific standards in description and presentation in the discussion part. I think that manuscript could be improved by some changes listed below:

1-The manuscript must be revised completely before it can be submitted and reviewed again. I would suggest to re-write its introductions, discussion and conclusion.

RESPONSE: We have revised and re-written the mentioned sections.

2-The language must be improved drastically. Some of the sentences are misleading and often not to understand. It recommended an English language revision since frequently the use of the grammar and the syntax results quite weird.

RESPONSE: We have corrected the grammar throughout the manuscript.

Round 2

Reviewer 4 Report

Dear Editor

Yes, it has been modified according to our suggestions.

I recommend the paper for publication.

Kindest regards

Author Response

Once again, thank you for taking time to check the revisions

Reviewer 5 Report

The authors have substantially revised the manuscript according to comments. The current form is acceptable and to be published in Antibiotics

Author Response

Thank you once again for taking time and checking the revised version